# Triprotic Ammonium Oleate Ionic Liquid Crystal Lubricant for Copper-Copper Friction and Wear Reduction

**María-Dolores Avilés** [ID], **Ana-Eva Jiménez \*** [ID], **Ramón Pamies** [ID], **Francisco-José Carrión-Vilches** and **María-Dolores Bermúdez**

Grupo de Ciencia de Materiales e Ingeniería Metalúrgica, Universidad Politécnica de Cartagena, Campus de la Muralla del Mar, 30202 Cartagena, Spain

**\*** Correspondence: anaeva.jimenez@upct.es

**Abstract:** The triprotic ammonium carboxylate ionic liquid crystal (2-hydroxyethyl)ammonium oleate (MO) has been studied as a neat lubricant and as a lubricant additive in two base oils, PAO6 and 100N. The lubricants have been used in commercially pure copper-OFHC copper balls on disk reciprocating sliding contact at room temperature. Neat MO presents a very good lubricating performance, with a friction coefficient of 0.06 and a wear rate of OFHC copper disk of $4.15 \times 10^{-7}$ mm$^3$/N·m. These results are, respectively, 94% and 98% lower than those obtained for PAO6, with similar reductions obtained with respect to 100N. MO has also been studied as an additive in 2wt.% proportion. The severe abrasive wear mechanism that takes place in the presence of neat base oils is reduced by the MO additive.

**Keywords:** copper; protic ionic liquid; lubrication; additive

## 1. Introduction

One of the fields of research that is receiving much attention is the use of ionic liquids [1–3] and nanoparticles [4,5] as lubricant additives in base oils. The need for improving the tribological performance of sliding systems, in particular copper-copper contacts, is especially relevant in such strategic fields as those of electric and transport applications. The present work constitutes a preliminary study of the feasibility of using a triprotic ammonium oleate lubricant in the reduction of coefficients of friction and wear rates in OFHC copper-commercially pure copper contacts.

Protic ionic liquids [2] are an economical, less toxic [3] alternative to heterocyclic or heteroatoms containing ionic liquids. In particular, fatty acid-derived species from renewable resources are currently receiving much attention [3,6–19].

(2-hydroxyethyl)ammonium oleate (MO; Figure 1) is an ionic liquid crystal with an ordered lamellar or micellar structure with an amphiphilic nature [20]. The existence of the highly hydrophobic long alkyl chains could favor the compatibility with non-polar lubricant fluids, while the interaction between carboxylate groups and ammonium cations gives rise to a hydrophilic unit which could interact with more polar fluids and with metallic surfaces. Moreover, the presence of three active hydrogen atoms in the cation could enhance the hydrogen bond formation. These were the reasons for choosing MO for the present study.

Protic ionic liquids have been previously studied in OFHC copper-OFHC copper lubrication by our research group [21]. The diprotic ionic liquid crystal bis(2-hydroxyethyl)ammonium oleate (DO), analogous to MO but with two hydroxyethyl substituents and two ammonium protons in the cation, showed poor tribological performance when used as a neat lubricant in copper-copper pin-on-disk configuration, compared with other protic ionic liquids with shorter alkyl chains or with two carboxylate groups in the anion.

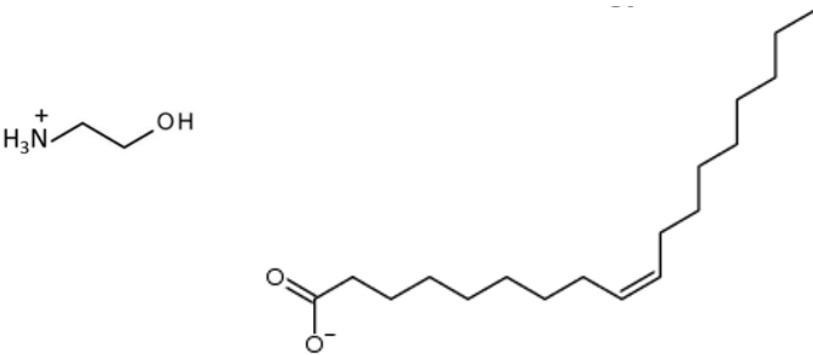

**Figure 1.** Molecular formula of 2-(hydroxyethyl)ammonium oleate (MO).

The main motivation of the present research is to improve the tribological performance of lubricant oils used in copper-copper contacts by their modification with a fatty acid derived as an environmentally friendly additive.

The presence of a third active proton in the ammonium cation of MO (Figure 1) could enhance the hydrogen bond formation ability and create a more effective lubricant film [16,22,23].

## 2. Materials and Methods

PAO6 has been previously described [21]. 100N is an isoparaffinic oil kindly supplied by Iberian Lube Base Oils Company, S.A. and has also been previously described [23]. (2-hydroxyethyl)ammonium oleate (MO) (Figure 1) was kindly supplied by M. Iglesias [20]. MO was added, in a 2 wt.% proportion, to PAO6 to obtain PAO6 + MO and to 100N to obtain 100N + MO, respectively.

Differential scanning calorimetry (DSC) studies were performed using a Mettler Toledo DSC-822e in the temperature range 0–150 °C at a heating rate of 10 °C/min under an $N_2$ atmosphere (50 mL).

Thermogravimetric analysis (TGA) was carried out with a TGA 1HT (Mettler Toledo) in the temperature range 30–600 °C, with a heating rate of 10 °C/min, in an $O_2$ (50 mL) atmosphere.

The rheological behavior of pure MO and of the base lubricants modified with MO was measured with an AR-G2 rotational rheometer from TA Instruments (New Castle, DE, USA). Due to the high viscosity and the stickiness and slippage of the neat MO, the experiments were performed with a hatched plate-plate configuration. In the case of the lubricants and the dispersions, a regular rotational plate with a diameter of 40 mm was used, and the gap between plates was set to 700 μm. A Peltier system was used for the control of the temperature with an accuracy of 0.1 °C. The steady-state viscosity was studied at room temperature by increasing the shear rate to 1000 s$^{-1}$.

Reciprocating tribological tests (Figure 2) were carried out at room temperature under ambient conditions in a TRB Anton Paar tribometer for a 44 m total sliding distance, under a normal load of 1.5 N, with a frequency of 2 Hz and a stroke distance of 5 mm.

Tribopair materials were commercially pure copper balls (Goodfellow, UK) with a diameter of 3 mm (<0.12 μm average roughness) and OFHC copper disks (99.95% Cu; 105 HV) disks (12.5 mm diameter 3.2 mm thickness; surface roughness Ra 0.08 μm). 1mL volume of lubricant was added to the surface of the copper disk before each test. In the case of neat MO, an orange-colored semi-solid, which is in the liquid crystalline state at room temperature, the amount of lubricant used was 0.005 g. All samples were cleaned with n-hexane and dried in the air before and after each test. Worn areas on pure copper balls were measured by image analysis using GIMP 2.10.30 software. Wear rates for OFHC copper disks were calculated from wear volumes as determined from cross-section areas along 4 mm wear track length (Figure 3) using Taylor Hobson Talysurf CLI 500 and Talymap software for the composition of 3D and 2D profiles. Tests were repeated at least three times to ensure reproducibility.

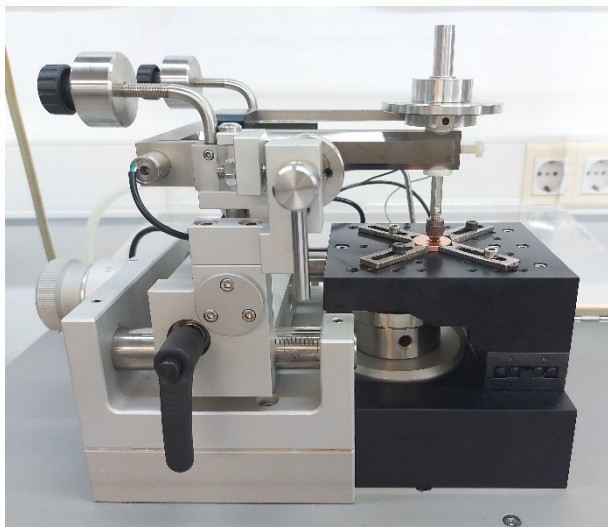

**Figure 2.** Set up reciprocating tribological tests.

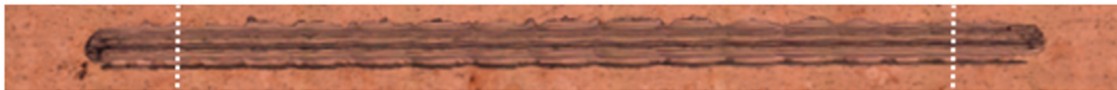

**Figure 3.** Wear track on OFHC copper disk showing the 4 mm length used for wear volume calculation.

Optical micrographs were obtained with a LEICA DMRX microscope. Scanning Electron Microscopy (SEM) micrographs and Energy Dispersive X-ray Spectroscopy (EDS) analysis were obtained with a FIB-SEM Crossbeam 350, Zeiss (Jena, Germany) and an Ultim Extreme EDS, Oxford Instruments (Abingdon, UK).

The order of experimental steps was the following: 1. Selection of base oils and ionic liquid additive. 2. Preparation of base oil + ionic liquid blends. 3. Thermal and rheological characterization. 4. Measurement of contact angles. 5. Set up materials and lubricants for tribological tests. 6. Continuous recording of friction coefficients. 7. Surface topography profile determination and wear volume measurement. 8. Observation of balls and disks surfaces by optical and electron microscopy. 9. EDX analysis. 10. Repetition of each test at least three times under the same conditions. 11. Calculation of average values and deviation.

## 3. Results

The lubricant blends PAO6 + MO and 100N + MO can be observed in Figure 4a,b, respectively. While MO is completely soluble in PAO6 (the pale yellow color of the PAO6 + MO blend is due to the orange color of MO), some turbidity can be appreciated in the case of 100N + MO. Nevertheless, both blends are stable for at least a period of several months. Differences in color and turbidity are attributed to different interactions between the liquid crystalline micelles of MO and the base oils.

Figure 5 shows DSC thermograms for all lubricants. Neat MO, which shows mesomorphic behavior, presents transitions at 6.9; 24.2; 37.0 and 102.0 °C; this shows that MO is in the mesomorphic state under the conditions used for tribological tests at room temperature. When MO is used as an additive in the base oils, these transitions are not observed, and only weak peaks appear at 43.9 °C for 100N + MO and at 46.20 °C for PAO6 + MO.

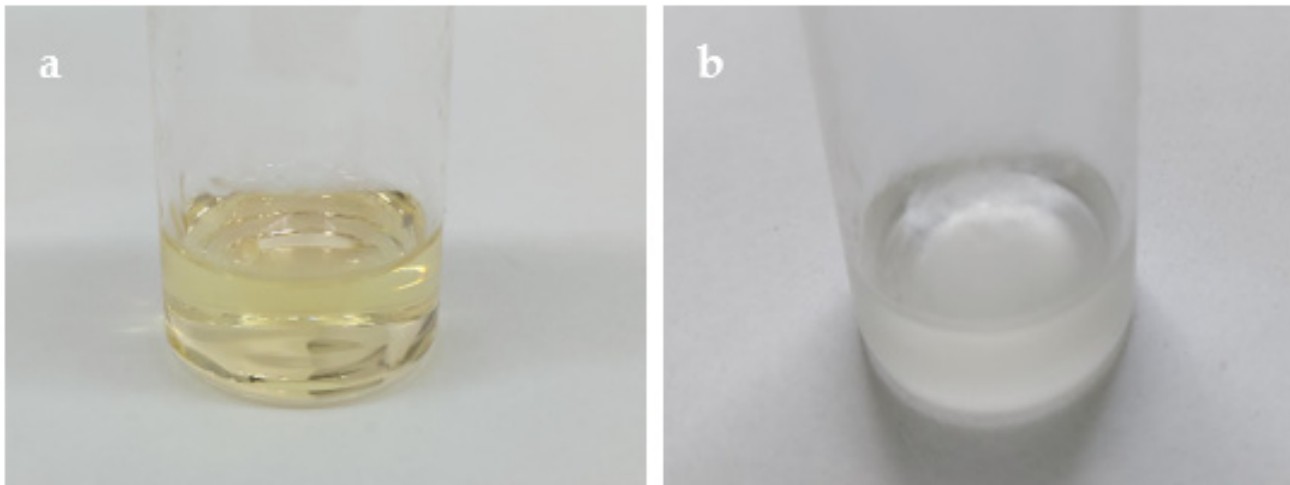

**Figure 4.** Photographs of the lubricant blends: (**a**) PAO6 + MO; (**b**) 100N + MO.

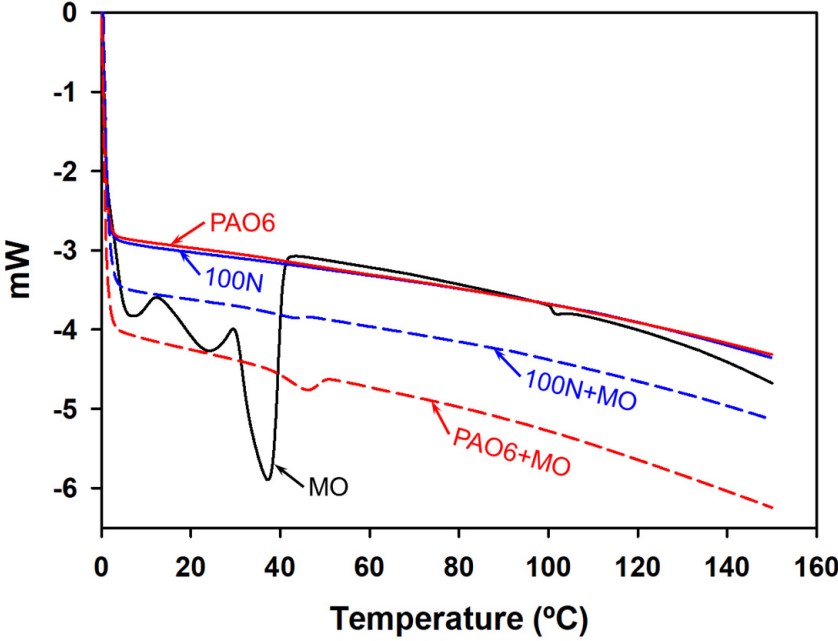

**Figure 5.** DSC thermograms of all lubricants.

Figure 6 and Table 1 present the results of the thermogravimetric analysis. The first mass loss step for neat MO, which starts at 62 °C and is centered at 90 °C, is probably due to the loss of absorbed water. However, the degradation temperature (50% weight loss) is higher for MO than for the base oils. In fact, the addition of MO increases the thermal stability of the base oils; this observation could be related to interactions of MO molecules with base oils. The highest thermal stability is observed for PAO6 + MO, where MO is completely soluble.

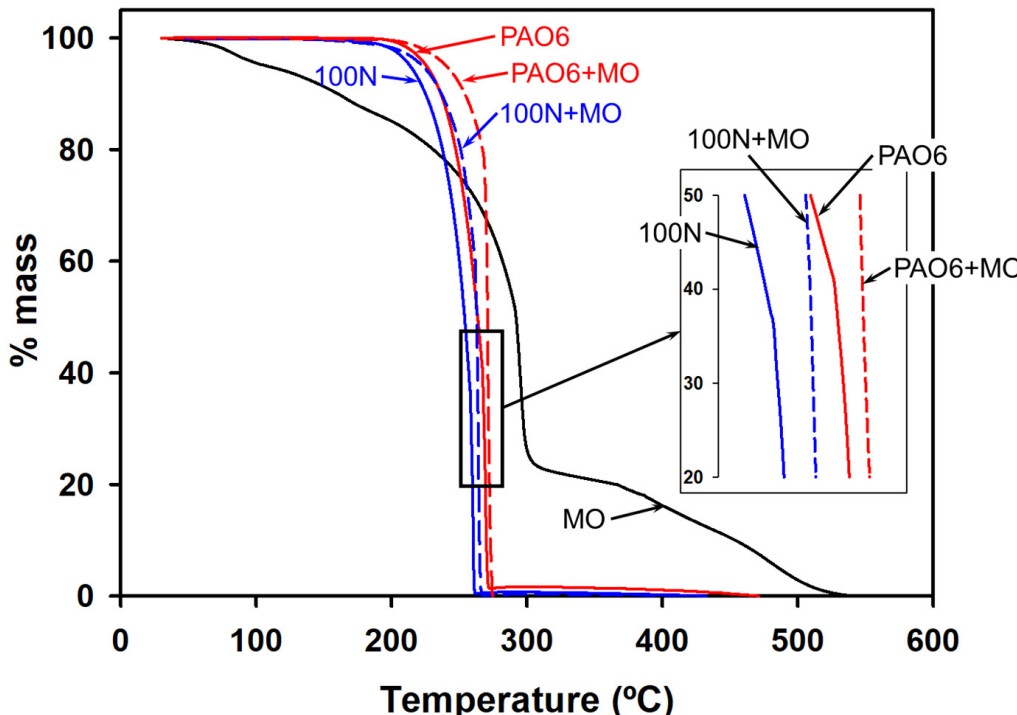

**Figure 6.** TGA curves for all lubricants.

**Table 1.** Thermal stability of the lubricants determined by TGA.

| Lubricant | T Onset (°C) | Td (−50 wt.%) * |
|---|---|---|
| MO | 62.16 | 312.3 |
| PAO6 | 258.0 | 284.8 |
| 100N | 246.9 | 270.5 |
| PAO6 + 2%MO | 288.0 | 379.7 |
| 100N + 2%MO | 274.7 | 330.7 |

* According to ASTM D-3850.

The rheological behavior of MO is shown in Figure 7a. As it can be seen, this ionic liquid crystal presents a strong shear thinning effect, with very high viscosity values at a low shear rate and a drop of the viscosity of 5 orders of magnitude. Conversely, 100N and PAO6 present a Newtonian behavior (Figure 7b). The viscosity values are 0.047 Pa·s for PAO6 and 0.030 Pa·s for 100N, respectively.

When MO is added to the lubricants, an increase in the viscosity can be observed, followed by a shear thinning effect. Probably, the ionic liquid crystal is present in the base lubricants in the form of big aggregates, and the increasing shear rate provokes the disruption of the agglomerates [10]. When MO is dispersed in 100N, a Newtonian plateau is reached at 100 s$^{-1}$, with a viscosity value of 0.036 Pa·s. However, this Newtonian region is not seen in PAO6 + MO due to stronger associations between the MO molecules.

Contact angles are presented in Table 2. Only initial and after 60 s values are shown, as no further changes were observed after this time. As can be observed, after just 60 s, the contact angles are very low for both base oils, thus showing the higher wettability on the OFHC surface. Changes in contact angles with the addition of MO could be indicative of direct interaction between MO molecules and the copper surface. The contact angle for neat MO could not be measured due to its semi-solid nature at room temperature.

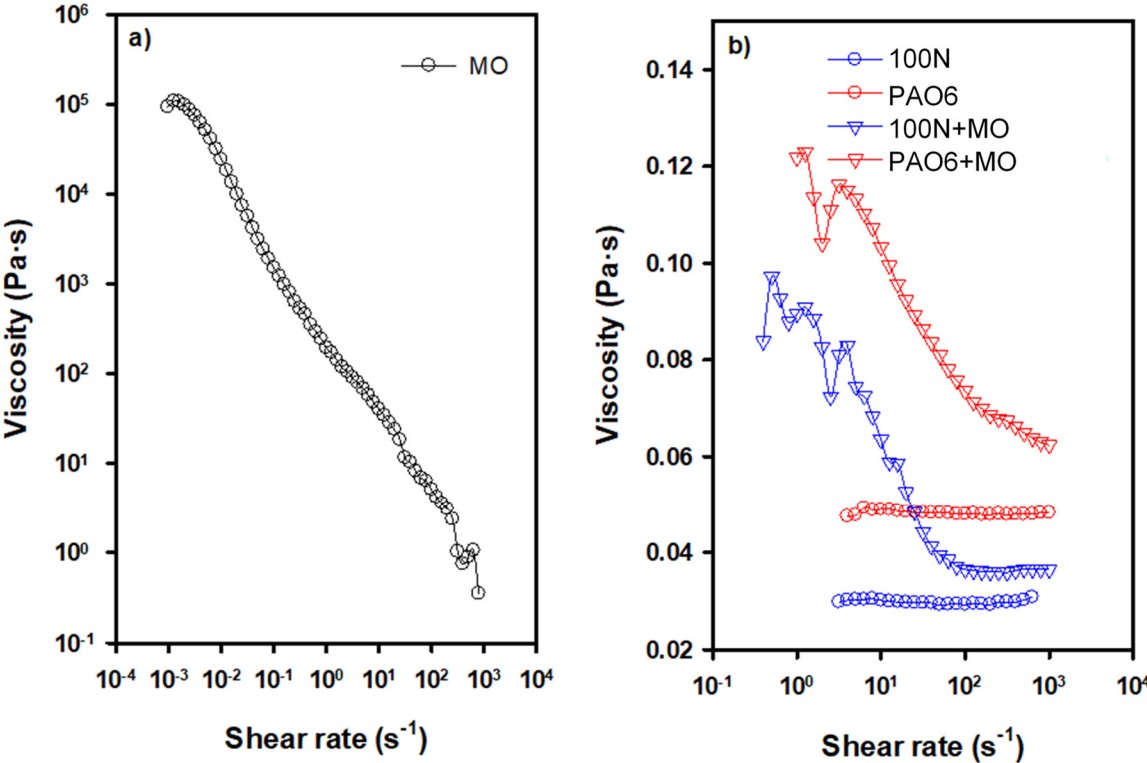

**Figure 7.** Influence of shear rate on viscosity: (**a**) MO; (**b**) PAO6, 100N, PAO6 + MO and 100N + MO.

**Table 2.** Contact angles on OFHC copper disk surface.

| Lubricant | Initial | After 60 s |
|---|---|---|
| PAO6 | 22.5° (±1.9) | 2.2° (±0.2) |
| 100N | 20.9° (±1.4) | 1.7° (±0.1) |
| PAO6 + MO | 27.6° (±1.1) | 17.1° (±1.5) |
| 100N + MO | 26.9° (±0.9) | 8.9° (±0.9) |

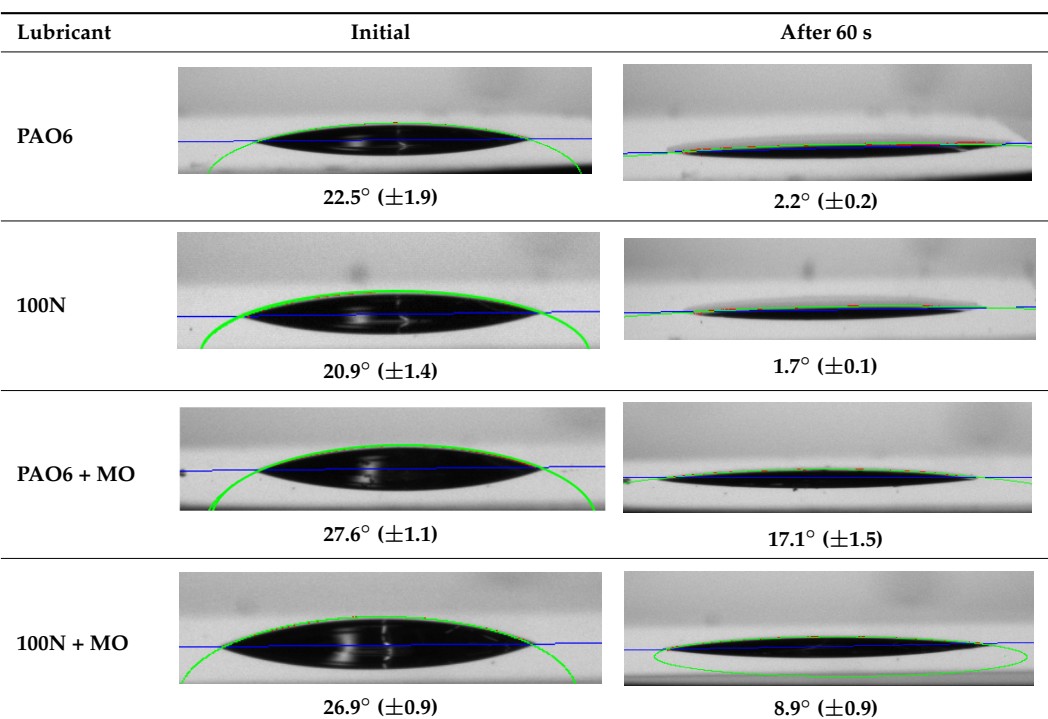

Figure 8 shows the evolution of the coefficients of friction with sliding distance for all lubricants.

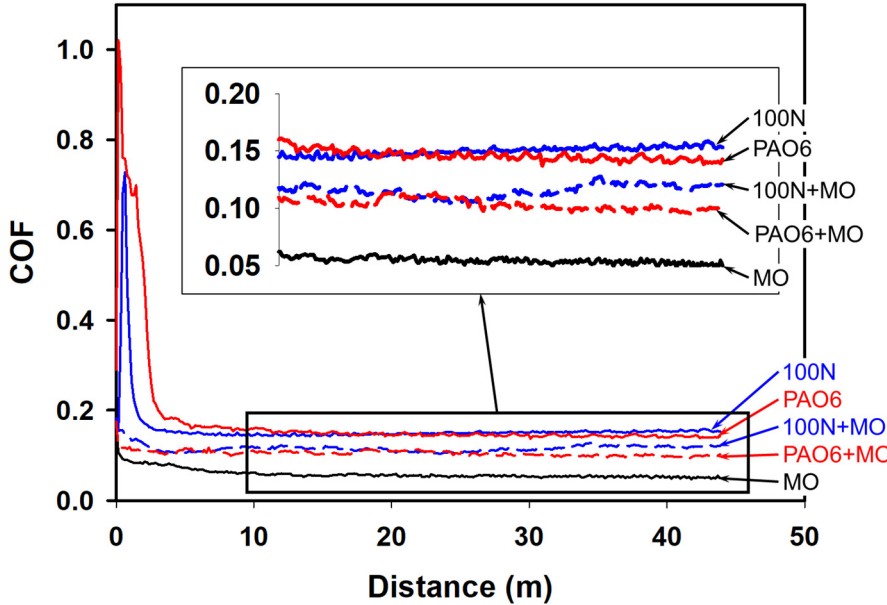

**Figure 8.** Variation of coefficients of friction (COF) with sliding distance for OFHC copper-pure copper sliding contacts in the presence of each of the lubricants and blends.

Table 3 shows friction values and wear rates for copper disks. Neat PAO6 and 100N lubricants show extremely high initial friction values. In both cases, a steady-state friction coefficient of 0.15 is reached after a short sliding distance.

**Table 3.** Coefficients of friction (COF).

| Lubricant | Maximum COF (Distance) | Steady-State COF | Average COF |
|---|---|---|---|
| MO | 0.16 (0 m) | 0.06 | 0.06 ($\pm$0.006) |
| PAO6 | 1.00 (0.17 m) | 0.15 | 0.18 ($\pm$0.010) |
| 100N | 0.73 (0.65 m) | 0.15 | 0.16 ($\pm$0.011) |
| PAO6 + 2%MO | 0.17 (0 m) | 0.10 | 0.10 ($\pm$0.002) |
| 100N + 2%MO | 0.18 (0 m) | 0.10 | 0.11 ($\pm$0.009) |

In contrast, neat MO shows a maximum initial friction coefficient of 0.16 to reach a steady-state value of 0.06 (Table 3) immediately. With an average friction coefficient of 0.06, the reduction of friction for MO with respect to PAO6 and 100N are 66% and 38%, respectively.

Thus, the use of MO as an additive of PAO6 and 100N presents several advantages with respect to the neat lubricants, such as the elimination of the initial high friction period and the reduction of the steady-state friction values (Table 3).

Neat MO lubricant shows a mild wear rate of $4.15 \times 10^{-7}$ mm$^3$/N·m, two orders of magnitude lower than those found for the base lubricant oils (Table 4). The wear-reducing ability of MO as a lubricant additive is also confirmed, with reductions of one order of magnitude (Table 4).

**Table 4.** Wear rate values for OFHC copper disks and worn areas on copper balls.

| Lubricant | Wear Rate (mm$^3$/N·m) | Worn Area (mm$^2$) |
|---|---|---|
| MO | $4.15 \times 10^{-7}$ ($\pm 3.76 \times 10^{-7}$) | $1.19 \times 10^{-2}$ |
| PAO6 | $2.50 \times 10^{-5}$ ($\pm 1.32 \times 10^{-5}$) | $1.10 \times 10^{-1}$ |
| 100N | $2.97 \times 10^{-5}$ ($\pm 5.27 \times 10^{-6}$) | $8.56 \times 10^{-2}$ |
| PAO6 + 2%MO | $5.08 \times 10^{-6}$ ($\pm 4.42 \times 10^{-6}$) | $2.57 \times 10^{-2}$ |
| 100N + 2%MO | $6.36 \times 10^{-6}$ ($\pm 1.67 \times 10^{-6}$) | $2.74 \times 10^{-2}$ |

Figures 9 and 10 show, respectively, cross-section profiles and surface topography of the wear tracks on OFHC copper disks. For comparative purposes, cross sections and wear tracks generated on different copper disks for each lubricant have been composed in one image. The cross-section profiles of the wear tracks on copper disks (Figure 9) show very mild wear for MO, severe abrasive surface damage for the base oils and mild abrasion for the lubricant blends containing MO additive. The total cross-section areas ($A_1 + A_2 + A_3$) were used for the calculation of wear rates.

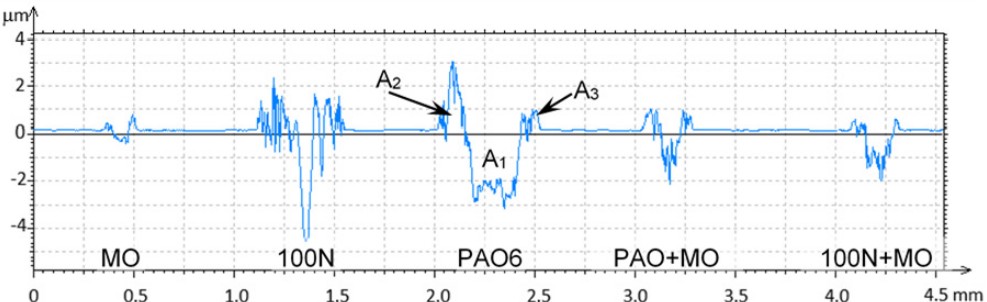

**Figure 9.** Cross-section profiles of the wear tracks on OFHC copper disks after tribological tests with each lubricant (2D profiles composition from different samples).

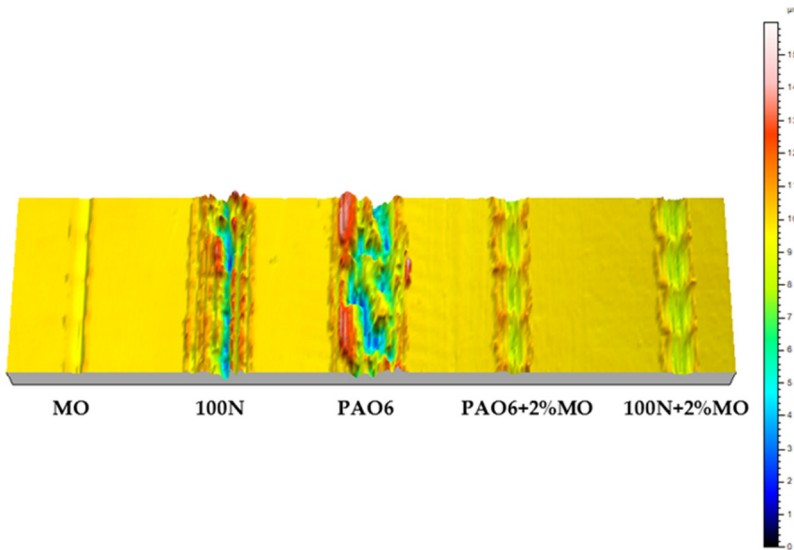

**Figure 10.** Surface topography 3D profiles of the wear tracks on OFHC copper disks (3D profiles composition from different samples).

The different wear mechanisms and surface damage on OFHC disks observed for each lubricant are confirmed in the surface topography 3D profiles shown in Figure 10.

Optical microscopy examination of the copper balls shows a similar trend.

Figure 11 shows optical micrographs of the copper balls after the tests. Worn area measurements have been shown in Table 4 and are in agreement with wear rates for OFHC copper disks. The wear mechanism for each lubricant is also confirmed by the observation of the worn areas for each lubricant. The very mild abrasion present in Figure 11a after lubrication with neat MO is transformed into very severe abrasion damage, with deep parallel grooves in the case of the base oils (Figure 11b,c); this very severe surface damage on the copper ball is reduced by the addition of MO to both base oils (Figure 11d,e); this reduction of surface damage severity is in agreement with previous results for OFHC copper disks sliding against OFHC copper pins under lubrication with PAO 6 modified by the addition of 1 wt.% the diprotic short alkyl chain ionic liquid di-[bis-(2-hydroxyethyl)ammonium] adipate [21].

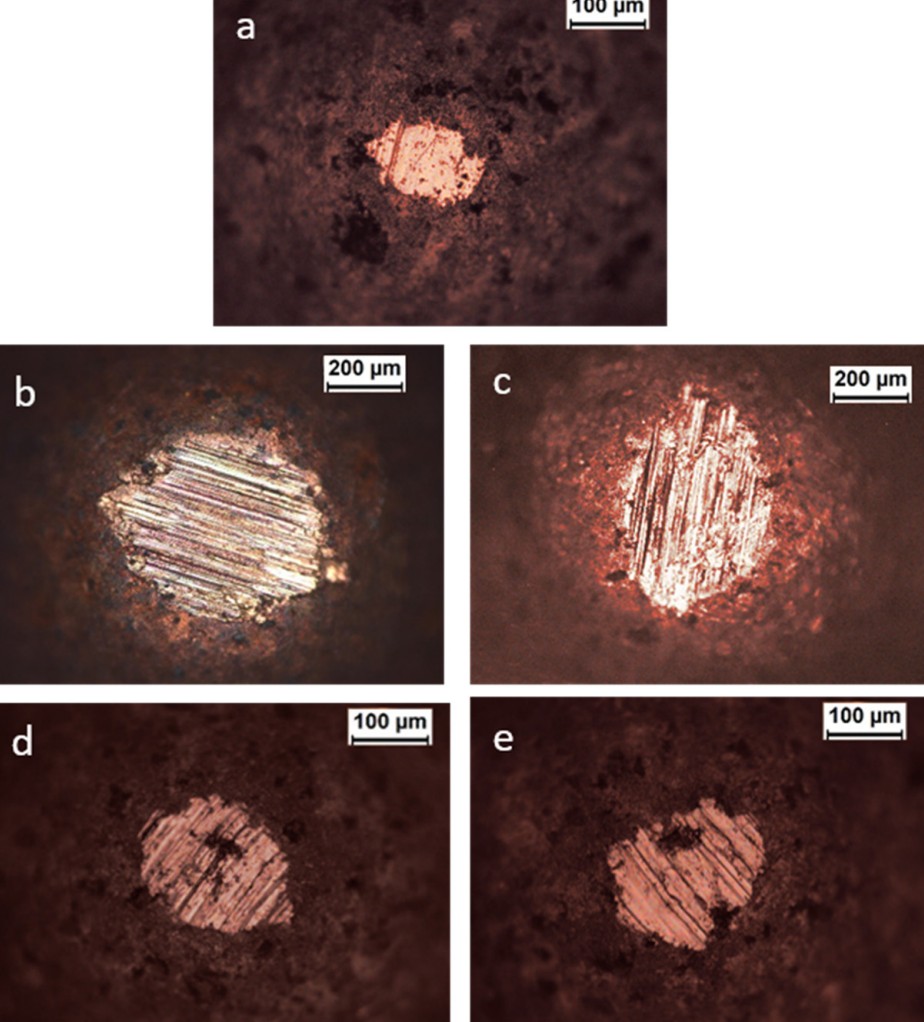

**Figure 11.** Optical micrographs of wear scars on copper balls after lubrication with: (**a**) MO; (**b**) PAO6; (**c**) 100N; (**d**) PAO6 + MO; (**e**) 100N + MO.

Wear tracks on OFHC copper disks were further studied by SEM and EDX. Figure 12a shows the wear track after lubrication with neat MO. Figure 12b corresponds to the EDX spectrum for spots 1, and 2 signaled in Figure 12a, inside and outside the wear track, respectively. Only copper is detected in both cases.

As we have seen in Figure 12, EDX spectra inside and outside the wear track on OFHC copper after lubrication with MO are the same. Table 5 shows the weight percentages of copper and carbon for the wear tracks after the tests with each of the lubricants. It is confirmed that there are no changes in the case of MO. In contrast, wear tracks after lubrication with neat base oils PAO6 and 100N show much lower copper contents and higher carbon concentrations, especially inside the wear tracks (Table 5). Figures 13 and 14 show that, with the addition of MO, wear tracks are narrower and less severe. Also, copper and carbon concentrations are again very similar inside and outside the wear tracks (Table 5); these results confirm the copper surface protection ability of MO. The high carbon percentages found for PAO6 and 100N could be related to residues after oil degradation.

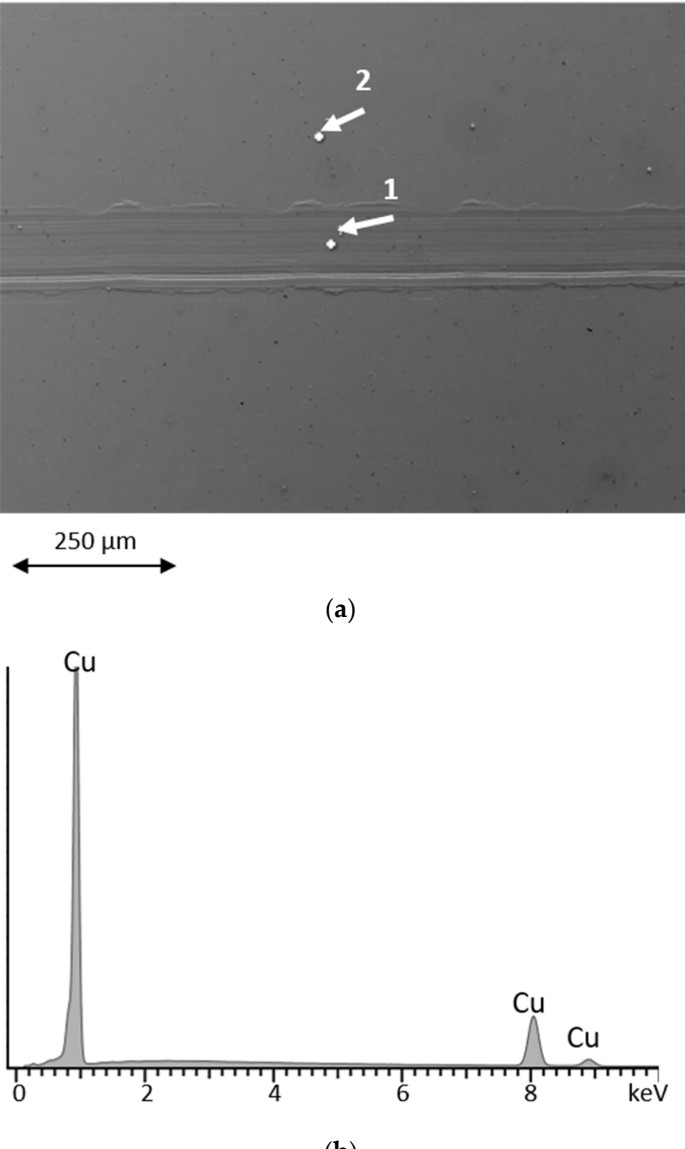

**Figure 12.** (**a**) SEM micrograph of the wear track on copper after lubrication with MO. (**b**) EDX spectrum inside (1) and outside (2) the wear track shown in (**a**).

**Table 5.** EDX results for the wear tracks on OFHC copper disks.

| Lubricant | Cu (wt.%) | | C (wt.%) | |
|:---:|:---:|:---:|:---:|:---:|
| | Inside | Outside | Inside | Outside |
| MO | 96.6 | 96.3 | 3.2 | 3.4 |
| PAO6 | 80.9 | 85.6 | 16.1 | 14.1 |
| 100N | 87.8 | 92.7 | 12.2 | 6.7 |
| PAO6 + 2%MO | 95.6 | 95.6 | 3.7 | 4.6 |
| 100N + 2%MO | 95.1 | 96.3 | 4.4 | 3.3 |

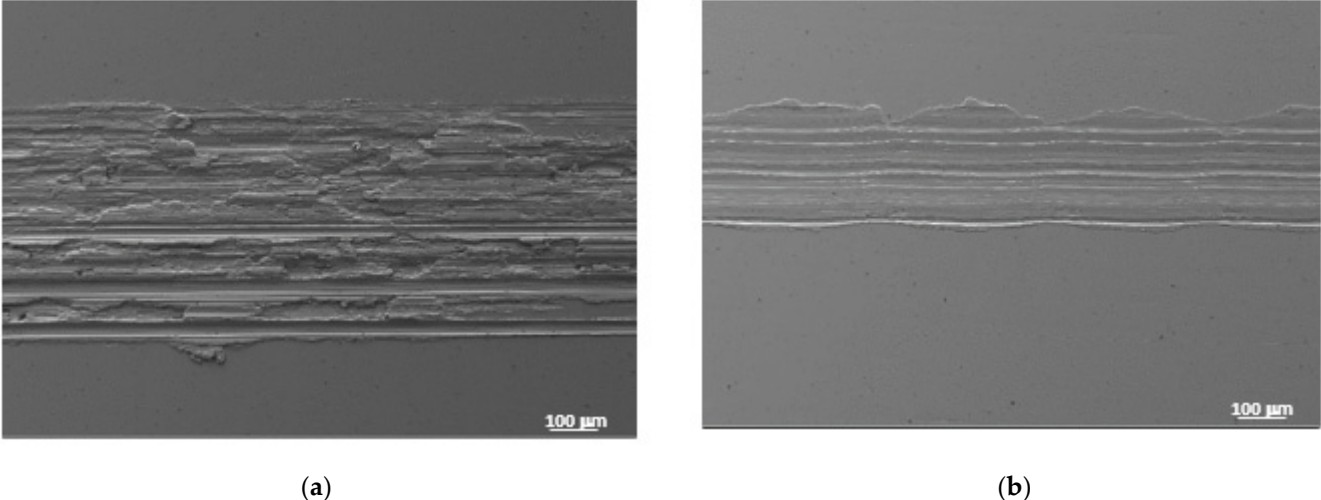

**Figure 13.** (**a**) SEM micrograph of the wear track on the copper disk after lubrication with PAO6.
(**b**) SEM micrograph of the wear track on the copper disk after lubrication with PAO6 + MO.

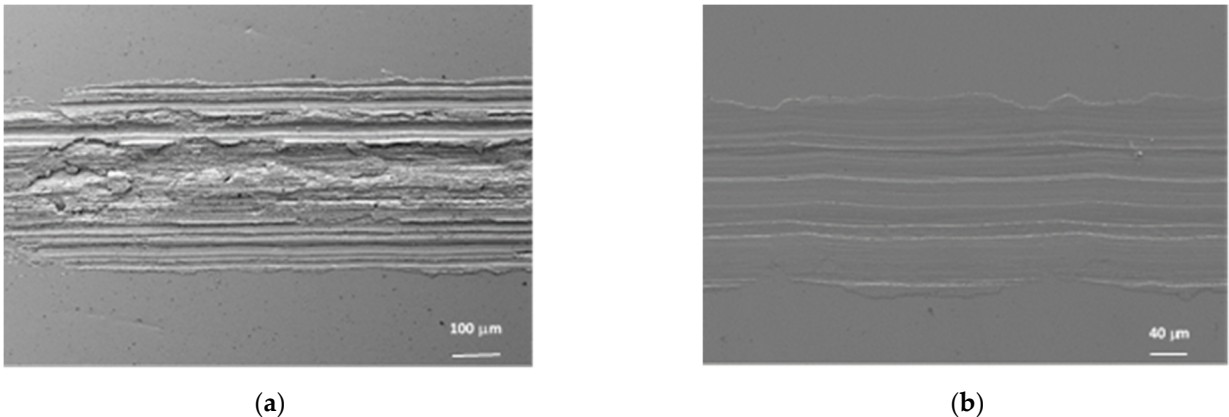

**Figure 14.** (**a**) SEM micrograph of the wear track on the copper disk after lubrication with 100N.
(**b**) SEM micrograph of the wear track on the copper disk after lubrication with 100N + MO.

The good tribological performance of MO is attributed to the formation of an easy shearing boundary layer, which also acts as an anti-wear and low-friction lubricant, reducing both friction and wear, particularly during the running-in period [24,25].

## 4. Conclusions

The triprotic ionic liquid crystal (2-hydroxyethyl)ammonium oleate has shown good tribological performance in copper-copper reciprocating contact, showing friction reduction and anti-wear ability both as a neat lubricant and as an additive in two synthetic oils.

This ionic liquid reduces both the running-in and the steady-state friction coefficient with respect to the base oils.

The very severe abrasive wear mechanism observed for the base oils is reduced to very mild surface damage for the neat ionic liquid.

This is the first time that this ionic liquid crystal described here has been used in the lubrication of copper-copper contacts, and the tribological results have been excellent, both as a neat lubricant and as a lubricant additive. The solubility of the additive in a commercial base oil such as PAO6 opens new practical applications of protic ionic liquids.

Further research will be focused on the study of the influence of MO concentration and temperature and on the analysis of surface interaction processes.

Advances in the understanding of the molecular interactions between base oils and additives also need to be addressed.

**Author Contributions:** Conceptualization, M.-D.A., A.-E.J. and M.-D.B.; methodology, M.-D.A., A.-E.J., R.P. and M.-D.B.; investigation, M.-D.A., R.P. and A.-E.J.; resources, M.-D.B.; writing—original draft preparation, M.-D.A.; writing—review and editing, M.-D.B.; funding acquisition, F.-J.C.-V. All authors have read and agreed to the published version of the manuscript.

**Funding:** This research was funded by MCIN/AEI/10.13039/501100011033/FEDER, UE. Grant number PID2021-122169NB-100.

**Data Availability Statement:** Not applicable.

**Conflicts of Interest:** The authors declare no conflict of interest.

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
