# Peer review of "Triprotic Ammonium Oleate Ionic Liquid Crystal Lubricant for Copper-Copper Friction and Wear Reduction"

_lubricants, doi:10.3390/lubricants10110290_

Round 1
Reviewer 1 Report
The experimental procedure is adequate, the article is well written, and the conclusions are supported by the results.
For this reason, I recommend its publication in Lubricants journal.
However, I recommend a few minor changes
Pag. 4. The authors indicate that: “Although the weight loss for MO starts around 100ºC, probably due to the loss of absorbed water”
However, in Fig. 5 it is observed that the loss of mass for MO begins remarkably below 100ºC. The curve representing the loss of mass has a more or less constant slope between approximately 70-230 ºC and an increase in slope around 100 ºC is not observed either. Consequently, this statement does not seem justified.
Since the MO begins its decomposition at a lower temperature than that of the base oils (Fig. 5). The authors should explain how it is possible that adding OM as an additive increases the decomposition start temperature of the mixtures.
What has been the method to determine the decomposition onset temperatures (T onset)? (see Table 1 and Fig. 5). The authors point out that the T onset of the MO is 292.3 ºC “Fort the highest mass los step”. But at that temperature there has already been a mass loss of about 80%, so it cannot be considered as the starting temperature of mass loss. From the analysis of Fig. 5, it does not seem justified to point out that the Td (-50 wt.%) is 312.3 ºC for MO.
Reviewer 2 Report
The authors have studied triprotic ammonium carboxylate ionic liquid crystal (2-hydroxyethyl)ammonium ole- 10 ate (MO) as neat lubricant and as lubricant additive in two base oils, PAO 6 and 100N. Suggestions are as follows:
1. The authors must give detailed information about the experimental setup, and the related pictures must be included.
2. The introduction section lacks sufficient descriptions for your motivation to do your work.
3. The authors should provide a detailed experimental flowchart if possible so that the proposed method can be referred by other researchers easily.
4. The novelty of the paper should be better stressed. It is not easy to understand what has been really achieved and what is new with respect to literature.
5. The authors should explicitly emphasize the major contributions of the paper.
6. The literature review might be updated by considering recent works published in a related fields.
a) https://doi.org/10.1016/j.apsusc.2013.02.083
b) https://doi.org/10.24874/ti.995.10.20.02
c) https://doi.org/10.1039/D0CS00126K
d) https://doi.org/10.1007/s11831-021-09538-1
7. Please add more punctual research directions that can be done from your paper.
8. Conclusions are not enough. It is better if you can indicate the wider applications of your techniques, emphasizing the global perspective of the problem they address. I suggest revising it according to the most interesting finishing of your paper.
9. Quality of the figures needs to be improved. It is difficult to interpret the results.
10. The repetition of the conducted and confirmation experiments is an important part that seems to be missing from the study.
11. Give literature citations in a spot of your claim of results. Analysis of the results obtained needs to be discussed in contrast to the previously carried out research.
12. To prove the novelty of this paper, a performance comparison to at least three related state-of-the-art works is needed.
13. Grammatical errors need to be addressed carefully.
Reviewer 3 Report
This study reported the tribological properties of (2-hydroxyethyl)ammonium oleate (MO) both as neat lubricant and as additive in two base oils.
My comments/questions are as follows:
1. It is better to add the photograph of MO to Fig.3. In addition, the lubricant blend PAO 6 + MO looks yellow. Is this caused by MO?
2. Sixty seconds may be too short to get a stable contact angle. Please provide a picture of the contact angle over time to prove that it doesn’t change much after 60 seconds.
3. According to Fig.8 and Fig.9, the tribological experiments of different lubricants seem to be carried out on the same OFHC copper disk. Does this contaminate surfaces?
4. The rheological behavior and wettability of lubricants are not reflected in the results section. The tribological mechanism of MO needs further discussion. Some papers can be used for reference:
https://doi.org/10.1016/j.carbon.2019.05.070
https://doi.org/10.3390/ma15155177
Round 2
Reviewer 3 Report
The authors fulfilled all my comments.